# High Sensitivity Fiber Interferometric Strain Sensors Based on Elongated Fiber Abrupt Tapers

**DOI:** 10.3390/mi13071015

**Published:** 2022-06-27

**Authors:** Haimiao Zhou, Ya-Pei Peng, Nan-Kuang Chen

**Affiliations:** 1School of Physics Sciences and Information Technology, Liaocheng University, Liaocheng 252000, China; 1910110302@stu.lcu.edu.cn; 2College of Engineering Physics, Shenzhen Technology University, Shenzhen 518000, China; 3NK Photonics Ltd., Jinan 250119, China

**Keywords:** few-mode interferometers, strain sensors, abrupt-tapered interferometers, elongated fibers, tensile strain

## Abstract

We demonstrate high-sensitivity fiber strain sensors based on an elongated abrupt taper. The fiber abrupt taper, with a tapered diameter ranging from 40–60 μm, was made by using a hydrogen microflame to break the waveguide adiabaticity so as to convert the fundamental mode into cladding modes. The abrupt taper was further uniformly tapered by using a normal moving flame with a torch diameter of 7 mm to elongate the tapered region until the tapered diameter was down to 2.5–5 μm. The excited high-order modes were confined to propagate along the cladding and then recombined at the rear edge of the fiber taper to produce interferences with extinction ratios of up to 16 dB. The tapered region was pulled outwardly to change the optical path difference (OPD) between modes to measure the tensile strain with all the interfering wavelengths blue-shifted. The measured best strain sensitivity was 116.21 pm/με and the coefficient of determination R2 of linear fitting exhibits high linearity. This strain sensor based on elongated abrupt taper is several times higher than that of most of the fiber strain sensors ever reported.

## 1. Introduction

In recent years, in-line fiber interferometer sensors are featured with high sensitivity, high accuracy, high environmental stability, and high signal capacity, and have been widely employed in many industrial and scientific applications [1,2,3,4]. Compared with the dual-fiber interferometer sensor, the in-line fiber interferometer sensor based on core mode and cladding mode interference is a more compact and efficient device, which is very suitable for industrial strain measurement including the health monitoring of bridge, propeller of the ships, and the blades of the wind turbines and aircraft engines [5,6,7,8,9,10]. The fiber strain sensors are usually made by using fiber Bragg gratings or fiber interferometers like Mach–Zehnder interferometers (MZI) [11,12,13,14,15,16,17]. The principle is that when the sensing region of fiber interferometer or grating is subjected to tensile or compressive stress, its resonant wavelength will accordingly change. Therefore, in order to improve the strain sensitivity, it is crucial to enlarge the optical path difference (OPD) between modes involving the interferences. On the other hand, a longer interaction length is also advantageous to improve the strain sensitivity. To date, there have been various kinds of excitation methods proposed to excite the high-order core or cladding modes based on blaze gratings, long-period gratings (LPFG), core mismatched splicing, fiber Bragg grating (FBG), abrupt tapering, up tapering [18,19], and so on. As a strain sensor, LPFG has high sensitivity and low back reflection, but it may block the spectral response due to its high bending sensitivity [20]. FBG must be limited by temperature-induced spectral shifts and isolators to prevent back reflection. In 2009, Tian et. al. reported dual-cone type MZI strain sensor with sensitivity of 2000 nm/ε [21]. However, dual-cone type MZI requires precise control of the distance between its two abrupt regions; otherwise good interference cannot be formed.

In this work, the fiber interferometer with high strain sensitivity based on a high efficiency mode conversion together with an OPD improving method is proposed. To excite the high-order cladding modes, the standard single-mode fiber SMF-28 was abruptly tapered to reach to a tapered diameter D_1_ to break the waveguide adiabaticity using a hydrogen micro torch. A fractional amount of the fundamental core mode was converted into cladding modes to propagate along the tapered fiber. The abrupt taper point was further elongated by another scanning flame torch to thin down the fiber to a wavelength scale of around 3 μm, defined as D_2_. Over the elongated tapered region, the mode fields distributed over the entire thinned cladding, which makes the tapered cladding turns out to be a new core. Under such circumstances, the air serves as the new cladding. Therefore, the excited high-order core modes propagate along the tapered fiber to produce interferences. Compared with the MZI strain sensor prepared based on two abrupt tapers, the prepared strain sensor has a simple structure, is easily manufactured, and has high stability and high sensitivity (higher than several times).

## 2. Experimental Setup

Figure 1 shows the experimental set-up for making high-strain sensitivity interferometers. The inset picture in Figure 1 shows that the microphotograph of the elongated abrupt taper under a (a) 50× and (b) 1000× CCD microscope, respectively. The SMF-28 was heated and abruptly tapered by using a hydrogen microflame to make an abrupt taper with D_1_ of around 40–60 μm, as shown in Figure 1a. The abrupt taper was further elongated by using a standard torch with a flame diameter of 7 mm to thin down the D_2_ until a few microns, shown in Figure 1b. A fiberoptic tensile strain sensor is fabricated, which consists of tapered optical fibers several microns in diameter with two abrupt tapered fibers at a bilateral junction between tapered and non-tapered regions. The abrupt tapered fiber structure can stimulate the propagation of higher-order modes along the microfiber, causing mode interference at the second tapered structure. Define the distance from the forming point of the first taper to the formation point of the second taper as L_1_. A broadband light source comprising superluminescent diodes (SLD) over a 1250–1650-nm wavelength range was used to measure the interferences of the interferometers. The wavelength shifts due to applying tensile strain were recorded by an optical spectrum analyzer (OSA) (YOKOGAWA AQ6370D, Tokyo, Japan). The bilateral edges of the elongated abrupt taper were fixed by using clampers to pull outward to measure the spectra transmitted by their tapered fibers versus the elongation of the tapered fiber and calculation of the tensile strain sensitivity. It is found that the tensile strain can be significantly improved when D_2_ is substantially decreased to a few microns. The tensile strain is calculated by the following equation.
∆ε = ∆L/L_2_, (1)
where △L and L_2_ respectively represent the elongation and the distance between two clampers.

## 3. Experimental Results and Analysis

In measurement of optical characteristics of the interferometers, four samples were prepared, with their corresponding working parameters shown in Table 1. We conducted stress-sensing experiments on four samples respectively. The D_1_ and D_2_ are ranging from 40–60 μm and 2.5–5 μm, respectively. The interferences of the interferometers from two of the samples, D_2_ = 2.62 μm and 4.49 μm, are shown in Figure 2. The optical resolution (RES) of the OSA was 0.05 nm. From Figure 2, the FSR obviously decreases with decreasing D_2_ because the OPD are increased. The achieved extinction ratios (ERs) are typically higher than 10 dB and can be up to 16 dB. The free spectral range (FSR) can be as narrow as 4.2 nm and decreases with decreasing tapered diameter D_2_. However, many unwanted modes can be removed; for D_2_ = 2.62 μm, oscillation curves of the interferences become clearer, compared with the ripples on the curve of D_2_ = 4.49 μm. The best strain sensitivity is 116.21 pm/με, several times higher than that of the most fiber strain sensors ever reported, with a highly linear response.

In measurement of strain sensitivity, the interferometers were respectively fixed by the two clampers with a distance L_2_ at the positions closing to the tapered region as possible as they can be. The two clampers were subsequently moved outward, and it would change by 2 μm per step to ensure the accuracy of the measurements. The data was recorded for 10 min after each movement, and the spectral responses were recorded by the OSA, as shown in Figure 3, Figure 4, Figure 5 and Figure 6. The spectral responses under different tensile strain range are provided only for the sample having the best strain sensitivity. From Figure 3, Figure 4, Figure 5 and Figure 6, it is found that the wavelength dips with a blue shift with an increasing tensile strain. In the spectral responses, we arbitrarily chose four wavelength dips, labeled ai, bi, ci, di, i = 1–4 for fitting calculation. The best strain sensitivity was calculated by linear fitting with a dip wavelength offset, shown in Figure 3, Figure 4, Figure 5 and Figure 6. The corresponding strain sensitivity can be calculated by using Equation (1), and those working parameters are listed in Table 1. The D_2_ of the samples are 4.97 μm, 4.52 μm, 4.49 μm, and 2.62 μm, where the corresponding strain sensitivity are 48.67 pm/με, 84.82 pm/με, 104.14 pm/με, and 116.21 pm/με, respectively. The coefficient of determination R^2^ of linear fitting exhibits high linearity with the corresponding R^2^ of 0.994, 0.987, 0.998, and 0.999, respectively. The best strain sensitivity is 116.21 pm/με and is several times higher than that of the most fiber strain sensors ever reported, using hollow-core fibers, photonic crystal fibers, multicore fibers, and multimode fibers [15,22]. We also carried out repeatability experiments and found that the sensor has high repeatability and good stability, as shown in Figure 7.

## 4. Conclusions

In conclusion, the high-sensitivity fiber interferometric strain sensors using the elongated abrupt taper were demonstrated. The abrupt taper was achieved by a microflame to convert the core mode into high-order cladding modes. The abrupt taper point was then heated and elongated until a tapered diameter of around 2.5–5 μm was achieved, so as to increase the OPD for good interferences and high strain sensitivity. The best ER is 16 dB and the best strain sensitivity is 116.21 pm/με with a high linearity in curve fitting. The interfering wavelengths blue-shifted with increasing tensile strain. The strain sensors based on the elongated abrupt-tapered interferometers having the strain sensitivity several times higher than that of the most fiber strain sensors ever reported. Compared with tapered multi-core fiber and fiber F-P strain sensor, and it has lower cost and higher sensitivity (higher than tens of times). Moreover, the tapered optical fiber interferometer still retains good mechanical properties of optical fiber. They are simple and cost-effective and are promising for developing micro-interferometric strain sensors with a small footprint. The micro/nano-strain sensors based on abrupt tapers can used in ultra-precision strain measurement applications, such as residual stress in thin films on microelectro mechanical system (MEMS) devices in the future. Moreover, the interferometer can be packaged after production to ensure that it can maintain good performance in harsh environments in practical applications. In the future, we need to think about how to package and protect the sensor while maintaining its sensitivity.

## Figures and Tables

**Figure 1 micromachines-13-01015-f001:**
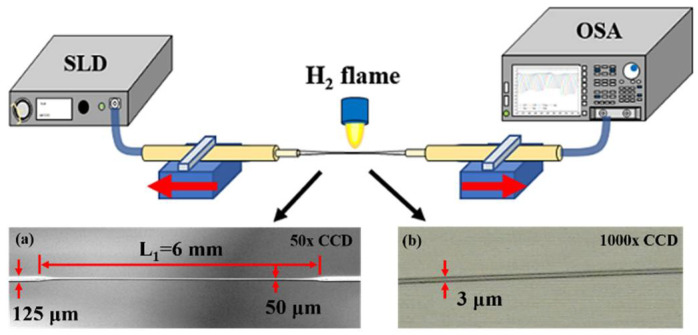
Experimental set-up of the strain sensor using an elongated abrupt taper;(**a**) microphotograph of a 50× CCD microscope and (**b**) microphotograph of a 1000× CCD microscope.

**Figure 2 micromachines-13-01015-f002:**
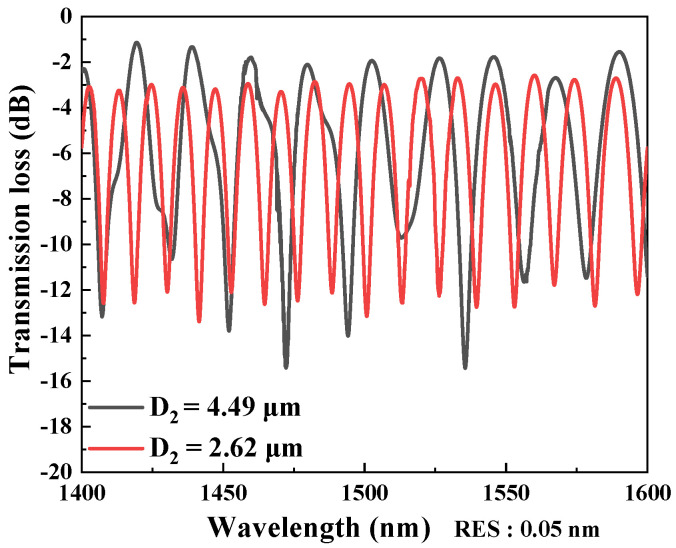
Spectral responses of the interferometers with D_2_ = 2.62 μm and 4.49 μm.

**Figure 3 micromachines-13-01015-f003:**
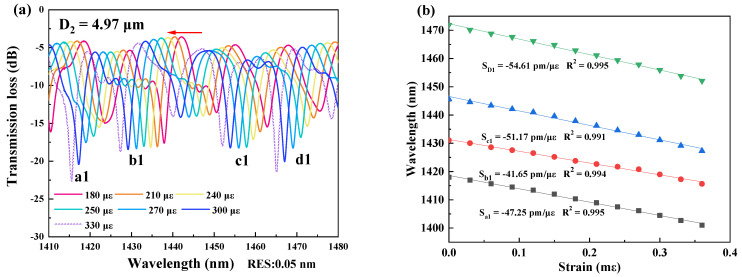
(**a**) Spectral responses of the interferometer with D_2_ = 4.97 μm in the strain range of 180–330 με at 1410–1480 nm. (**b**) Linear fitting curves of the dip wavelength shifts and the coefficients of determination R^2^ of linear fitting with D_2_ = 4.97 μm.

**Figure 4 micromachines-13-01015-f004:**
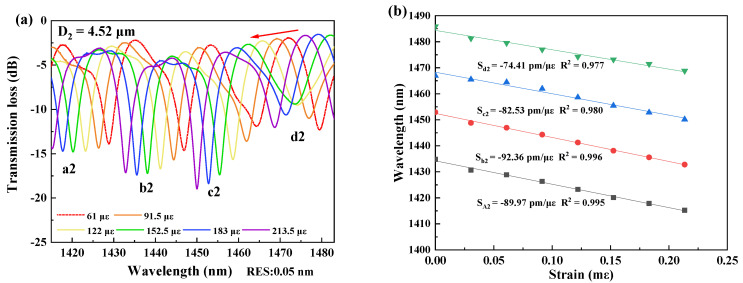
(**a**) Spectral responses of the interferometer with D_2_ = 4.52 μm at the strain ranging over 61–213.5 με at 1415–1483 nm. (**b**) Linear fitting curves of the dip wavelength shifts and the coefficients of determination R^2^ of linear fitting with D_2_ = 4.52 μm.

**Figure 5 micromachines-13-01015-f005:**
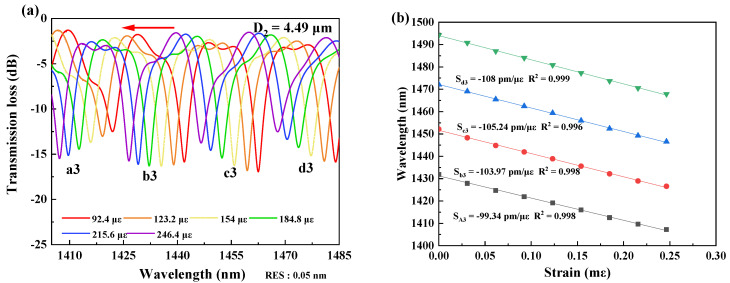
(**a**) Spectral responses of the interferometer with D_2_ = 4.49 μm at the strain ranging over 92.4–246.4 με at 1400–1485 nm. (**b**) Linear fitting curves of the dip wavelength shifts and the coefficients of determination R^2^ of linear fitting with D_2_ = 4.49 μm.

**Figure 6 micromachines-13-01015-f006:**
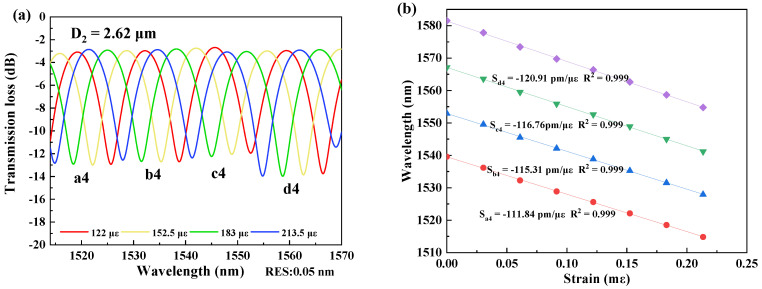
(**a**) Spectral responses of the interferometer with D_2_ = 2.62 μm at the strain ranging over 122–213.5 με at 1513–1570 nm. (**b**) Linear fitting curves of the dip wavelength shifts and the coefficients of determination R^2^ of linear fitting with D_2_ = 2.62 μm.

**Figure 7 micromachines-13-01015-f007:**
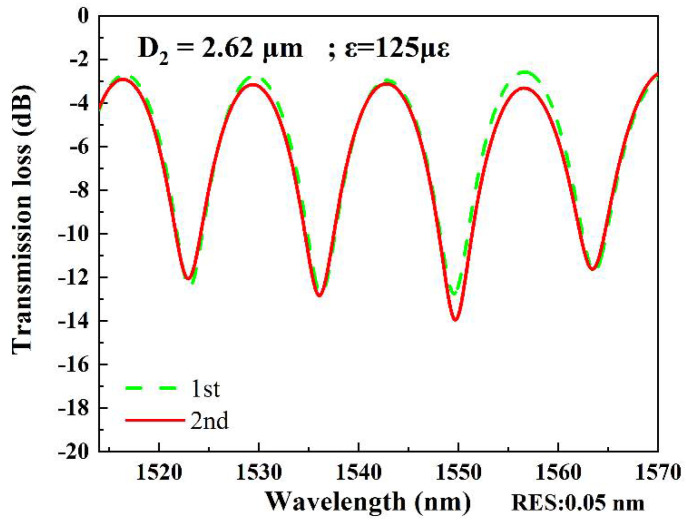
The repeatability experiments on sample with D_2_ = 2.62 um.

**Table 1 micromachines-13-01015-t001:** Working parameters of the strain sensors.

Parameters	Sample1	Sample2	Sample3	Sample4
Tapered diameter D_1_	4.97 μm	4.52 μm	4.49 μm	2.62 μm
Clamper distance L_2_	6.7 cm	6.55 cm	6.5 cm	6.55 cm
Strain Sensitivity $\left(\bar{S}\right)$	48.67 pm/με	82.84 pm/με	104.14 pm/με	116.217 m/με
Strain (με)	0~330 με	0~213.5 με	0~246.4 με	0~213.5 με
Coeff. Of determination R^2^	0.994	0.987	0.998	0.999

## Data Availability

The data that support the findings of this study are available upon request from the corresponding author. The data are not publicly available due to privacy or ethical restrictions.

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
