# Peer review of "High Sensitivity Fiber Interferometric Strain Sensors Based on Elongated Fiber Abrupt Tapers"

_micromachines, 2022, doi:10.3390/mi13071015_

Round 1

Reviewer 1 Report

According to my recommendations before publication in the MDPI "Micromachines," the authors must correct the paper.

1. The introduction part is not satisfactory. What is missing is a detailed review of other solutions in the literature.

2. Lines 42 - 49 and lines 64-67 are duplicates of the same information. The authors should decide where to place information about the preparation of the sample.

3. Line 86. Table 1 should be placed below the sentence corresponding to this table, not at the end of the manuscript.

4. Fig. 2. in my opinion, is hardly unreadable. The picture quality is very low, and the authors should correct this. The red dashed curve could be replaced with a solid line. 

5. Fig. 3a. Too many curves in the picture make the chart hard to analyze. Also, I recommend using a larger gamut of colors (not different hues of the blue color). The same concerns involve Fig.  4a, 5a, and 6a.

6. In the conclusion part, a comparison between these results and other solutions from literature would be desired. 

Author Response

According to my recommendations before publication in the MDPI "Micromachines," the authors must correct the paper.

  1. The introduction part is not satisfactory. What is missing is a detailed review of other solutions in the literature.

Thanks for the reviewer's suggestion. We had added some review of reported literature on the introduction part.

  1. Lines 42 - 49 and lines 64-67 are duplicates of the same information. The authors should decide where to place information about the preparation of the sample.

Thanks for the reviewer's reminder. We had revised it.

  1. Line 86. Table 1 should be placed below the sentence corresponding to this table, not at the end of the manuscript.

Thanks for the reviewer's suggestion. We had revised it.

  1. Fig. 2. in my opinion, is hardly unreadable. The picture quality is very low, and the authors should correct this. The red dashed curve could be replaced with a solid line.

Thanks for the reviewer's suggestion. We had revised it.

  1. Fig. 3a. Too many curves in the picture make the chart hard to analyze. Also, I recommend using a larger gamut of colors (not different hues of the blue color). The same concerns involve Fig.  4a, 5a, and 6a.

Thanks for the reviewer's suggestion. We had revised it.

Fig 3(a) and fig 4 (a)

Fig. 5(a) and Fig. 6(a)

  1. In the conclusion part, a comparison between these results and other solutions from literature would be desired.

Thanks for the reviewer's suggestion. We had revised it.

Reviewer 2 Report

May, 25, 2022
Review on the paper (Micromachines_1760360)
"High Sensitivity Fiber Interferometric Strain Sensors based on Elongated Fiber
Abrupt Tapers"
by Haimiao Zhou et al.
This paper reports on a Mach-Zehnder interferometer (MZI) founded on a tapered single
mode fiber, for application as a strain sensor. The principle of this MZI has been well-known
for many years, and a large number of papers have already been reported on sensors based on
such MZI in the past (see for example the review paper by Korposh et al. (
Sensors 2019, 19,
2294; doi:10.3390/s19102294) and the one by
Taha et al. (Biosensors 2021, 11, 253.
https://doi.org/10.3390/bios11080253)).
However, most MZI in tapered fibers are used as refractive index sensors, whereas, in the current
paper, the MZI is used as an elongation sensor. The second notable point is the high sensitivity of the
best of the reported sensors which is higher than 100 pm/µ
.
Comments section by section
1- Introduction
2- In this introduction part, applications of strain sensors cited by the authors (monitoring health
of bridges, propeller of ships, blades of wind turbines, …) require robust devices, since they
must work in harsh environment. Then, sensors based on tapered fibers are likely to be easily
breakable. For that reason, the authors should give examples and references showing practical
applications of such sensors in industrial environment.
3- Typo in ref 4 : “tapered fiber-optic” instead of “tapered fifiber-optic”.
4- Line 38 : meaning of OPD (=optical path difference) should be explicitly given when this
abbreviation is used for the first time.
5- Line 53 : the sentence “Generally, the extinction ratios….” reports performances which are
those of the sensors which will be depicted in the paper. Thus, it should be transferred in the
discussion part or in the conclusion.
6- Line 55: the best sensitivity of 118.77 pm/µ
is not in accordance with the highest value
reported in the paper (116.21 pm/µ
). The authors should correct this discrepancy.
7- Line 57: I don’t understand what the authors mean by “the MZI strain sensor prepared by
optical fiber”. What do they mean?
8- Lines 59 to 61. This last sentence is not clear. The authors should more clearly explain why
the distance of the MZI sensor is critical. Is it due to the coherence length of the source? At
least, they should provide suitable references on this point.
2- Experimental setup
9- Line 66: the microphotograph in Fig. 1 seems to not really represent the actual profile of the
taper. Indeed, the length L1 of 3.19 mm measures 3.5mm in this picture, i.e. the scale ratio
(actual length)/(length on the picture) is 1.1. With this scale ratio, the diameter D2 of 2.62µm
should measure 2.9µm in the picture, too thin for being visible in the figure. Likewise, the
diameter of the non-tapered fiber (125µm) should measure about 0.14mm in the figure
whereas it actually measures about 1mm. The authors should clarify this point. They should
display a profile of their taper with a larger scale.
10- Line 74: Is the light from the SLD polarized? Is the spectrum measured for a given strain on a
given sample sensitive to the polarization of the incident light? For example, is this spectrum
sensitive to curvatures and/or twists applied to the fiber before the taper? This point should be

discussed because, if the peak wavelengths are shifted depending on the polarization, this
could have a harmful influence on the precision of the measurements.
11- The Fig.2 is too small for being suitably exploitable. Furthermore, the dashed lines (for D2_2)
are not suitable for properly showing the corresponding spectrum (readability), even with
enlarged scale on the computer screen. The authors should carefully improve these curves.
12- Line 77 and following: actually, the authors measure the spectra transmitted by their tapered
fibers versus the ELONGATION of the taper and not versus the applied strain, which is not
known. Even if this confusion is often found in the literature, the authors should add a warning
sentence for the readers.
13- Line 81 : the quantity
L is called “displacement”. I think it would be better to call it
“elongation”. The authors should consider this point.
14- Line 90 and following: how do the authors measure the diameter of their tapers (from 4.97µm
to 2.62µm)? How can they guarantee their measures with such a precision of 10
-2µm ? Is the
diameter constant all along the central region between the abrupt tapers?
15- Line 98 and following: the authors report strain sensitivity for their different samples with a
precision of 10
-2pm/µwhich is not realistic. They should evaluate the uncertainty of their
measurement and they should indicate it in their experimental results.
16- Fig. 3a : A1, B1, C1 and D1 should be defined (I guess they correspond to the different peaks
identified in the spectrum).
17- Fig.3: D1 is used for identifying a peak in the spectrum whereas it is also used for the diameter
of the abrupt taper. Likewise, in Fig.4, D2 is used for identifying a peak in the spectrum
whereas it is also used for the diameter of the tapered fiber. The authors should correct these
confusing denominations.
18- Fig 3b : this figure seems to be simply a zoom on the range [1450nm 1475nm] of Fig 3a, for
13 spectra measured from 0µ
to 330µ. . In this case I don’t understand the strain ranges
indicated in the caption : “(a)…..strain ranging over 0-210µ
, and (b) 240-330µ” . The
authors should clarify this point. Same remarks for Fig. 4 and Fig. 5.
19- Fig 4: The spectrum measured for the strain of 213.5µ
exhibits a peak at about 1484nm
(green dashed line), i.e. between the peak at 1486nm for the strain of 0µ
(black dashed line)
and the peak at 1482nm for the strain of 32.5µ
(red line). This means that the only
measurement of this peak could be interpreted as the signature of a strain between 0 and 32.5
µ
. In other words, there is an ambiguity due to the multiple peaks in one spectrum. To my
mind, this ambiguity could be removed only by means of a continuous monitoring of one
spotted peak, starting from the reference spectrum for the strain of 0µ
. There are other similar
confusing situations in the different figures. The authors should carefully consider and
comment this issue. They should also reduce accordingly the range of measurable strains, if
necessary.
20- What about the repeatability of the measures? Have the authors made repeated measures with
the same sensor (i.e. elongation, then back to the starting conditions, and same elongation
again) ? The authors should comment on this point.
21- Line 126: table1 How do the authors define the working range of their samples in terms of
strain (for example 0
330µfor sample 1)? Is the upper limit defined when the sensor
breaks?
22- Several typos: line 88 “40-60 µm” instead of “40-60 @µm”. Line 111: “116.21pm/µ

instead of“116.21pm/@
”. Lines 115, 118, 121, 124 “fitting” instead of “fit-ting”.
3. Conclusion

23- Due to their extreme narrowness (few µm), the taper must be particularly fragile. Thus the
authors should comment on the practical implementation of such sensors. In particular, it is
well-known that silica glass is made friable and brittle by environment humidity. What about
the packaging which should be conceived for protecting the sensor while preserving the
sensitivity ? In what domain and for what particular application such a sensor could be useful
and performing ?
In conclusion, this paper reports on highly sensitive strain sensors based on tapered fibers. However
different issues and points to be clarified have been pointed out in this review. On the other hand, the
reported experimental results should be of better interest if the authors could identify specific
applications requiring such sensitive but fragile sensors. All the points listed above, should be
carefully addressed for making this paper suitable for publication in “Micromachines”. Major revision
is required.

Author Response

May, 25, 2022
Review on the paper (Micromachines_1760360)
"High Sensitivity Fiber Interferometric Strain Sensors based on Elongated Fiber
Abrupt Tapers"
by Haimiao Zhou et al.
This paper reports on a Mach-Zehnder interferometer (MZI) founded on a tapered single mode fiber, for application as a strain sensor. The principle of this MZI has been well-known for many years, and a large number of papers have already been reported on sensors based on such MZI in the past (see for example the review paper by Korposh et al. (Sensors 2019, 19, 2294; doi:10.3390/s19102294) and the one by Taha et al. (Biosensors 2021, 11, 253. https://doi.org/10.3390/bios11080253)).
However, most MZI in tapered fibers are used as refractive index sensors, whereas, in the current paper, the MZI is used as an elongation sensor. The second notable point is the high sensitivity of the best of the reported sensors which is higher than 100 pm/µe.
Comments section by section
1- Introduction

2- In this introduction part, applications of strain sensors cited by the authors (monitoring health of bridges, propeller of ships, blades of wind turbines, …) require robust devices, since they must work in harsh environment. Then, sensors based on tapered fibers are likely to be easily breakable. For that reason, the authors should give examples and references showing practical applications of such sensors in industrial environment.

As we all know, optical fiber can be free from magnetic field interference and atomic radiation, adapt to harsh environment with soft quality, small size, light weight, high sensitivity of mechanical resistance to high temperature, chemical resistance to corrosion, etc. The tapered optical fiber interferometer still has good mechanical properties. And the interferometer is packaged after production to ensure that it can maintain good performance in harsh environments for practical applications.

3- Typo in ref 4 : “tapered fiber-optic” instead of “tapered fifiber-optic”.

Thanks for the reviewer's reminder. We had revised it.

4- Line 38 : meaning of OPD (=optical path difference) should be explicitly given when this abbreviation is used for the first time.

We had revised it.

5- Line 53 : the sentence “Generally, the extinction ratios….” reports performances which are those of the sensors which will be depicted in the paper. Thus, it should be transferred in the discussion part or in the conclusion.

Thanks for the reviewer's suggestion. We had revised it.

6- Line 55: the best sensitivity of 118.77 pm/µe is not in accordance with the highest value reported in the paper (116.21 pm/µe). The authors should correct this discrepancy.

Thanks for the reviewer's reminder. We had revised it.

7- Line 57: I don’t understand what the authors mean by “the MZI strain sensor prepared by optical fiber”. What do they mean?

We mean the MZI strain sensor prepared based on two abrupt tapers type. We had revised it to avoid misunderstanding.

8- Lines 59 to 61. This last sentence is not clear. The authors should more clearly explain why the distance of the MZI sensor is critical. Is it due to the coherence length of the source? At least, they should provide suitable references on this point.

The two optical paths of the biconical MZI must have a certain optical path difference, otherwise interference cannot be formed.  Because the coherence length of white light is very limited (on the micron scale), the optical paths of all wavelengths of white light must be very carefully tuned, and there must be a tightly controlled value to form interference. Compared with this, the single-cone optical fiber interferometer developed by us is easier to control.

2- Experimental setup
9- Line 66: the microphotograph in Fig. 1 seems to not really represent the actual profile of the taper. Indeed, the length L1 of 3.19 mm measures 3.5mm in this picture, i.e. the scale ratio (actual length)/(length on the picture) is 1.1. With this scale ratio, the diameter D2 of 2.62µm should measure 2.9µm in the picture, too thin for being visible in the figure. Likewise, the diameter of the non-tapered fiber (125µm) should measure about 0.14mm in the figure whereas it actually measures about 1mm. The authors should clarify this point. They should display a profile of their taper with a larger scale.

Thanks for the reviewer's reminder. We had revised the fig.1.

2
10- Line 74: Is the light from the SLD polarized? Is the spectrum measured for a given strain on a given sample sensitive to the polarization of the incident light? For example, is this spectrum sensitive to curvatures and/or twists applied to the fiber before the taper? This point should be discussed because, if the peak wavelengths are shifted depending on the polarization, this could have a harmful influence on the precision of the measurements.

SLD is an incoherent broadband light source with wide wavelength coverage and high power stability. SLD can be regarded as unpolarized light.

11- The Fig.2 is too small for being suitably exploitable. Furthermore, the dashed lines (for D2_2) are not suitable for properly showing the corresponding spectrum (readability), even with enlarged scale on the computer screen. The authors should carefully improve these curves.

Thanks for the reviewer's suggestion. We had revised it.

12- Line 77 and following: actually, the authors measure the spectra transmitted by their tapered fibers versus the ELONGATION of the taper and not versus the applied strain, which is not known. Even if this confusion is often found in the literature, the authors should add a warning sentence for the readers.

Thanks for the reviewer's suggestion. We had revised it.

“The bilateral edges of the elongated abrupt taper were fixed using clampers for pulling outward to measure the spectra transmitted by their tapered fibers versus the elonga-tion of the taper fiber and calculation the tensile strain sensitivity”

13- Line 81 : the quantity DL is called “displacement”. I think it would be better to call it “elongation”. The authors should consider this point.

Thanks for the reviewer's suggestion. We had revised it.

14- Line 90 and following: how do the authors measure the diameter of their tapers (from 4.97µm to 2.62µm)? How can they guarantee their measures with such a precision of 10-2µm ? Is the diameter constant all along the central region between the abrupt tapers?

Optical fiber diameter is measured by 1000x microscope, which can accurately measure optical fiber with a diameter of several microns. The 1000X microscope can accurately read values to two decimal places. The diameter of the tapered fiber is constant when viewed under a 1000X microscope.

15- Line 98 and following: the authors report strain sensitivity for their different samples with a precision of 10-2pm/µe which is not realistic. They should evaluate the uncertainty of their measurement and they should indicate it in their experimental results.

The sensitivity was obtained by linear fitting of sampling points by ORIGN software, and two decimal places were reserved when we read the sensitivity value. The R2 values obtained by the fitting are basically above 0.99, indicating that our fitting results are correct.

16- Fig. 3a : A1, B1, C1 and D1 should be defined (I guess they correspond to the different peaks identified in the spectrum).

We had added some defined in the manuscript: “In the spectral responses, we arbitrarily chose four wavelength dips, labeled ai, bi, ci, di, i =1-4 for fitting calculation. The best strain sensitivity was calculated by linear fitting with dip wavelength offset, shown in figs. 3-6(b).”

17- Fig.3: D1 is used for identifying a peak in the spectrum whereas it is also used for the diameter of the abrupt taper. Likewise, in Fig.4, D2 is used for identifying a peak in the spectrum whereas it is also used for the diameter of the tapered fiber. The authors should correct these confusing denominations.

Thanks for the reviewer's reminder. We had revised it.

18- Fig 3b : this figure seems to be simply a zoom on the range [1450nm 1475nm] of Fig 3a, for 13 spectra measured from 0µe to 330µe. . In this case I don’t understand the strain ranges indicated in the caption : “(a)…..strain ranging over 0-210µe, and (b) 240-330µe” . The authors should clarify this point. Same remarks for Fig. 4 and Fig. 5.

We had revised it.

19- Fig 4: The spectrum measured for the strain of 213.5µe exhibits a peak at about 1484nm (green dashed line), i.e. between the peak at 1486nm for the strain of 0µe (black dashed line) and the peak at 1482nm for the strain of 32.5µe (red line). This means that the only measurement of this peak could be interpreted as the signature of a strain between 0 and 32.5 µe. In other words, there is an ambiguity due to the multiple peaks in one spectrum. To my mind, this ambiguity could be removed only by means of a continuous monitoring of one spotted peak, starting from the reference spectrum for the strain of 0µe. There are other similar confusing situations in the different figures. The authors should carefully consider and comment this issue. They should also reduce accordingly the range of measurable strains, if necessary.

We have adjusted the color of the spectral lines to make them better for observation. Large strain ranges with different diameters are selected for the accuracy of experimental results, while small strain ranges may be interfered by external factors that lead to inaccurate experimental results.

20- What about the repeatability of the measures? Have the authors made repeated measures with the same sensor (i.e. elongation, then back to the starting conditions, and same elongation again) ? The authors should comment on this point.

We also carried out repeatability experiments and found that the sensor has high repeatability and good stability.

3
21- Line 126: table1 How do the authors define the working range of their samples in terms of strain (for example 0 ~ 330µe for sample 1)? Is the upper limit defined when the sensor breaks?

The strain measurement range in Table 1 is gradually explored through experiments. However, due to the later need to measure the stability and repeatability of the sensor, the upper limit of stress was not determined.

22- Several typos: line 88 “40-60 µm” instead of “40-60 @µm”. Line 111: “116.21pm/µe” instead of“116.21pm/@e”. Lines 115, 118, 121, 124 “fitting” instead of “fit-ting”.

We had revised it.

  1. Conclusion
    23- Due to their extreme narrowness (few µm), the taper must be particularly fragile. Thus the authors should comment on the practical implementation of such sensors. In particular, it is well-known that silica glass is made friable and brittle by environment humidity. What about the packaging which should be conceived for protecting the sensor while preserving the sensitivity ? In what domain and for what particular application such a sensor could be useful and performing ?

We had added some comments in the conclusion.

In conclusion, this paper reports on highly sensitive strain sensors based on tapered fibers. However, different issues and points to be clarified have been pointed out in this review. On the other hand, the reported experimental results should be of better interest if the authors could identify specific applications requiring such sensitive but fragile sensors. All the points listed above, should be carefully addressed for making this paper suitable for publication in “Micromachines”. Major revision is required.

Round 2

Reviewer 1 Report

All of my concerns have been addressed properly.

I recommend publication of the manuscript.

Author Response

Thank you for reviewer's comments.

Reviewer 2 Report

June, 14, 2022
Review on the revised paper (Micromachines_1760360)
"High Sensitivity Fiber Interferometric Strain Sensors based on Elongated Fiber
Abrupt Tapers"
by Haimiao Zhou et al.
In the revised version of the above paper, most of the requirements expressed in the review
concerning the initial version are taken into consideration. In particular, the figures have been
significantly improved, which was absolutely necessary.
However, two main weak points remain in this revised version.
First, the authors claim that their sensor exhibits high stability (line 68). They also state that
they carried out repeatability experiments and found that the sensor has high repeatability and
good stability (lines 170-171). However, they do not report any experimental result
confirming these affirmations. What is considered as “high repeatability”? To support these
statements, several curves (at least 2 or 3) showing the spectral response of one given
interferometer versus strain along repeated cycles of strain from 0 to a maximum value
(before breaking!) should have been convincing.
The second weak point is the kind of application for which such a sensor should be useful,
even necessary. Such application is not explicitly cited. As expressed at the end of the
conclusion, the packaging of such a sensor, achieving mechanical protection while preserving
high sensitivity is obviously a serious issue.
Notwithstanding these two remarks, this revised version of the paper could be considered as
acceptable for publication in Micromachines.

Author Response

Review on the revised paper (Micromachines_1760360) "High Sensitivity Fiber Interferometric Strain Sensors based on Elongated Fiber Abrupt Tapers" by Haimiao Zhou et al. In the revised version of the above paper, most of the requirements expressed in the review concerning the initial version are taken into consideration. In particular, the figures have been significantly improved, which was absolutely necessary. However, two main weak points remain in this revised version. First, the authors claim that their sensor exhibits high stability (line 68). They also state that they carried out repeatability experiments and found that the sensor has high repeatability and good stability (lines 170-171). However, they do not report any experimental result confirming these affirmations. What is considered as “high repeatability”? To support these statements, several curves (at least 2 or 3) showing the spectral response of one given interferometer versus strain along repeated cycles of strain from 0 to a maximum value (before breaking!) should have been convincing. The repeatability experiments on sample D2=2.62 um is shown in fig.1. Fig.1 The second weak point is the kind of application for which such a sensor should be useful, even necessary. Such application is not explicitly cited. As expressed at the end of the conclusion, the packaging of such a sensor, achieving mechanical protection while preserving high sensitivity is obviously a serious issue. Thank you for reviewer’s comments. The micro/nano sensors based on abrupt tapers prepared are mainly used in ultra-precision strain measurement applications, such as residual stress in thin films. The properties of microscopic materials are quite different from those of macroscopic materials [1]. The film residual stress generated during the process is an important factor that must be considered in the design and fabrication of MEMS devices. Therefore, we prepared the micro/nano strain sensors is beneficial to the development of micro electro mechanical system (MEMS) technology. However, the mechanical protection problem is an engineering problem, and we'll work on that later. [1] S. P. Murarka and T. F. Retajczyk Jr. “Effect of phosphorus doping on stress in silicon and polycrystalline silicon “ (1983) J. Appl. Phys. 54 2069 Notwithstanding these two remarks, this revised version of the paper could be considered as acceptable for publication in Micromachines.
